# Renalase Overexpression-Mediated Excessive Metabolism of Peripheral Dopamine, DOPAL Accumulation, and α-Synuclein Aggregation in Baroreflex Afferents Contribute to Neuronal Degeneration and Autonomic Dysfunction

**DOI:** 10.3390/biomedicines13051243

**Published:** 2025-05-20

**Authors:** Xue Xiong, Yin-Zhi Xu, Yan Zhang, Hong-Fei Zhang, Tian-Min Dou, Xing-Yu Li, Zhao-Yuan Xu, Chang-Peng Cui, Xue-Lian Li, Bai-Yan Li

**Affiliations:** State Key Laboratory of Frigid Zone Cardiovascular Diseases (SKLFZCD), Department of Pharmacology (State Key Laboratory-Province Key Laboratories of Biomedicine-Pharmaceutics of China, Key Laboratory of Cardiovascular Research, Ministry of Education), College of Pharmacy, Harbin Medical University, Harbin 150086, China; xuexiong@buffalo.edu (X.X.); xu2019@purdue.edu (Y.-Z.X.); zhangy0393@163.com (Y.Z.); 15145227962@163.com (H.-F.Z.); 18712028460@163.com (T.-M.D.); 15097657976@163.com (X.-Y.L.); xu1990@purdue.edu (Z.-Y.X.); 18845146616@163.com (C.-P.C.); lixuelian@hrbmu.edu.cn (X.-L.L.)

**Keywords:** renalase, dopamine, α-Synuclein, baroreflex afferents, autonomic failure, Parkinson’s disease

## Abstract

**Background/Objectives**: Increasing evidence reveals the likely peripheral etiology of Parkinson’s disease; however, the mechanistic insight into α-Synuclein aggregation in the periphery remains unclear. This study aimed to explore the effect of abnormal expression of renalase on dopamine metabolism, toxic DOPAL generation, and subsequently, α-Synuclein aggregation. **Methods**: Blood pressure (BP) was monitored while changing the body position of rats; the serum level of renalase was detected by ELISA; the mRNA/protein of renalase and α-Synuclein were determined by qRT-PCR/Western blot; DOPAL was measured using HPLC; renalase distribution was explored by immunostaining; cell viability and ultrastructure were examined by TUNEL and electron microscopy, respectively. **Results**: The results showed that, in PD model rats, the serum level of renalase was increased time-dependently with up-regulated renalase gene/protein expression in the nodose ganglia, nucleus tractus solitarius, and heart; a reduced dopamine content was also detected by the renalase overexpression in PC12 cells. Strikingly, up-regulated renalase and orthostatic BP changes were observed before the behavioral changes in the model rats. Meanwhile, the levels of DOPAL and α-Synuclein were increased time-dependently. Intriguingly, the low molecular weight of α-Synuclein declined coordinately with the increase in the higher molecular weight of α-Synuclein. Clear ultrastructure damage at the cellular level supported the notion of molecular findings. Notably, the α-Synuclein aggregation-induced impairment of the axonal transport function predates neuronal degeneration mediated by renalase overexpression. **Conclusions**: Our results demonstrate that abnormal peripheral dopamine metabolism mediated by overexpressed renalase promotes the DOPAL-induced α-Synuclein and leads to baroreflex afferent neuronal degeneration and early autonomic failure.

## 1. Introduction

Parkinson’s disease (PD) is a systemic disorder with motor prodromal symptoms and non-motor symptoms, including orthostatic BP changes, in which non-motor symptoms often appear earlier than motor disorders [1,2,3]. PD is characterized by the degeneration of dopaminergic neurons in the substantia nigra pars compacta with the aforementioned typical symptoms [4]. The classic theory holds that PD is caused by the loss of central dopaminergic neurons, resulting in movement disorders. However, studies have found that α-Synuclein (α-Syn) can misfold in the Peripheral Nervous System (PNS) to form oligomers and spread to the central nervous system (CNS) in a prion-like manner, leading to PD’s pathology [5,6]. α-Syn is a protein of only 14.5 kDa and is mainly localized within the presynaptic terminals, axon, and cell body of neurons. Although the causative agent of PD is still unclear, numerous studies have pinpointed α-Syn as a key factor. Dopaminergic neuron loss is associated with the abnormal accumulation and aggregation of α-Syn and formation of Lewy bodies (LB), suggesting thatα-Syn plays a key role in the neurodegenerative process of PD; however, the true mechanism of α-Syn aggregation remains debated and several lines of existing evidence have pointed out the crucial role of α-Syn: (1) the transcription factor NURR1 involved in the differentiation and maintenance of dopaminergic phenotypes is down-regulated by α-Syn [7]; (2) α-Syn aggregation activates tyrosine kinase c-Abl, which in turn promotes α-Syn aggregation, a feed-forward interaction [8,9]; (3) the disruption of dopaminergic signaling has been documented in α-Syn overexpression models showing the abnormal firing of dopaminergic neurons, leading to PD symptoms [10]; and (4) α-Syn at or near the surface membrane enhances the interaction with the DA transporter, consequently altering the conductance of the DA transporter, excitability of dopaminergic neurons, and thus DA neurotransmission [11]. Existing evidence has also shown that α-Syn could diffuse to the brain by injecting pathological α-Syn fibers into the duodenum and pyloric muscle layer of mice, leading to dopaminergic neuronal degeneration [12,13,14]. Clinical data have demonstrated that abnormal α-Syn aggregation could be detected in the enteric nervous system of PD patients [13,15]. Also, vagotomy and appendectomy could reduce the risk of PD [16]. Additionally, α-Syn can specifically damage dopaminergic neurons in the brain following transmission from the gut to the brain through the vagus nerve [17,18], highly suggesting that the origin of PD should be initiated from the periphery [19].

In the context of the catecholamine hypothesis on the development of PD and other LB diseases, attention has been increasingly given to 3,4-dihydroxyphenylacetaldehyde (DOPAL), which is catalyzed by amine oxidase (MAO) from DA in the synapse. DOPAL is an endogenous toxin detoxified by aldehyde dehydrogenase (ALDH) to 3,4- DOPAC and mediates the aggregation and quinonelation of multiple proteins including α-Syn, leading to death of dopaminergic neurons [20,21]. The catecholamine hypothesis also suggests that the DOPAL-induced α-Syn oligomerization impairs axon transport [22] and sequential DA neuronal degeneration [23,24,25].

RNLS was first identified in 2005 as a protein originally found in the kidneys with an endocrine function [26], while its expression can be detected in almost all organ systems. Initially studied for its potential amine oxidase activity due to its FAD-binding domain, RNLS is now recognized as a NADH/NADPH oxidase (EC 1.6.3.5) with broader catalytic activities [27]. While early studies suggested a role in catecholamine metabolism [28], its precise mechanisms and substrate specificity, particularly concerning direct DA oxidation, are still under investigation. Nevertheless, alterations in RNLS levels have been linked to changes in catecholamine homeostasis. This is supported by findings in RNLS knockout mice, which showed elevated DA levels that could be reversed upon the administration of recombinant RNLS [29]. These observations suggest that RNLS may regulate DA availability, potentially through its oxidase activity, which could influence related metabolic pathways or the cellular redox environment.

Taking all existing evidence together with our previous data, we hypothesize that the etiology of PD may be initiated due to DOPAL-mediated α-Syn abnormal aggregation in the peripheral autonomic nervous system. To this end, we overexpressed RNLS in PC12 cells and observed its impact on DA levels, leading to the time-dependent accumulation of DOPAL and α-Syn aggregation in this cellular model. Collectively, these results anticipate more insightful hints to comprehend PD etiology and early non-motor symptoms as well as pathological biomarkers, and will have significant clinical impacts on the diagnosis and management of PD.

## 2. Materials and Methods

### 2.1. Chemicals

DOPAL (3,4-dihydroxy-benzeneacetaldehyde) was purchased from Cayman Chemical Company (Ann Arbor, MI, USA). The stock solution was kept at −20 °C and diluted before experiments.

### 2.2. Experimental Animals

Male Sprague Dawley (SD) rats weighing 200–250 g were purchased from the experimental animal center of the Second Affiliated Hospital of Harbin Medical University (Harbin, China; the certificate number: SCXK-2019-001). All protocols for animals used in these experiments were approved by the Institutional Animal Care and Use Committee (IRB3030922 approved on 20 December 2022) of Harbin Medical University, which are in accordance with the recommendations of the Panel on Euthanasia of the American Veterinary Medical Association and the National Institutes of Health publication “Guide for the Care and Use of Laboratory Animals” (https://www.nap.edu/readingroom/books/labrats/, accessed on 14 January 2021).

### 2.3. 6-OHDA-Induced PD Model

Twenty rats were randomly divided into two groups: a sham group and experimental group (Experimental scheme, Appendix A). Rats were anesthetized using sodium isobarbital solution (25 mg/kg) after randomization. After fixing the head of the rat, the subcutaneous tissue was surgically opened, and the pericardium of the skull was bluntly separated. Referring to a standard rat brain stereotaxic atlas, unilateral (left side) injections were targeted at two sites: the substantia nigra pars compacta (SNc) and the ventral tegmental area (VTA). For SNc, the stereotaxic coordinates were as follows: Anteroposterior (AP), −5.0 ± 0.2 mm from bregma; Mediolateral (ML), −1.7 ± 0.1 mm from the sagittal suture; and Dorsoventral, (DV) −7.6 ± 0.1 mm from the skull surface. For VTA, the stereotaxic coordinates were as follows: AP, −4.6 ± 0.1 mm from bregma; ML, −0.9 ± 0.1 mm from the sagittal suture; and DV, −7.5 ± 0.1 mm from the skull surface. The wounds were sutured and injected with penicillin after operation. For the sham control rats, the same amount of normal saline was injected under the same condition. A dental drill was used to create small burr holes through the skull at the determined coordinates. A microsyringe was slowly advanced to the predetermined depth for each target site. For the PD model group, 6-hydroxydopamine (6-OHDA; Sigma-Aldrich, St. Louis, MO, USA) was injected. The 8-µg 6-OHDA dose (in a volume of 4 µL) was injected equally into two sites (SNc and VTA) at an injection rate of 1.0 µL/min. After each injection, the needle was left in place for 10 min to allow for diffusion and minimize backflow, and then slowly withdrawn. For the sham-operated control group, an equivalent volume of 0.9% sterile saline containing 0.02% ascorbic acid was injected into the same coordinates using the identical procedure. Following injections, the scalp wound was sutured routinely. To prevent infection, all rats received intraperitoneal injections of penicillin G Procaine at a dosage of 50,000 units/kg daily for one week post-surgery.

### 2.4. Behavioral Examination

Starting from one week after modeling, a fixed time was selected every week to perform a “rotation test” on the rats. The DA receptor agonist, apomorphine, was injected at a dose of 0.5 μg/g to observe whether the animals showed abnormal behavior such as rotation to the intact side or whether there were symptoms such as shaking and bradykinesia. If the rotation to the healthy side was greater than 7 circles per minute or greater than 210 circles per 0.5 h, the modeling was considered successful.

### 2.5. BP Measurements While Changing Position

To understand the effect of position (orthostatic or supine) on cardiovascular parameters, the mean artery pressure (MAP) and heart rate (HR) as well as baroreflex sensitivity (BRS) were collected under both the orthostatic and supine position of anesthetic rats before and after 6-OHDA application.

### 2.6. Elisa Kit Detection of Serum RNLS

The abdominal cavity of rat was exposed until light red blood vessels were visible. After taking the required amount of blood, the tube was placed at 4 °C overnight to allow stratification and centrifuged at 1000× *g* for 20 min. If hemolysis occurs, subsequent experiments cannot be performed. Each sample can only be frozen and thawed once and must be centrifuged before adding the sample. The concentration of RNLS was detected using the Rat RNLS ELISA Kit (CSB-EL002965RA, Cusabio, Wuhan, China) and the O.D. value was measured at 450 nm.

### 2.7. Tissue Collection

After complete relaxation with sodium amobarbital solution (3%, 1.5 mL/kg), 10 rats were sacrificed, and tissue of the nodose ganglion (NG) and nucleus tractus solitarius (NTS) were carefully dissected according the procedures described previously [30] and stored at −80 °C for subsequent molecular testing.

### 2.8. Cell Culture and Transfection

PC12 cells were purchased from Procell Life Science & Technology Co., Ltd. (CL-0481; Wuhan, China) and cultured in a 1640 medium (SH30096, Hyclone, UT, USA) supplemented with 10% fetal bovine serum (FB15015, Clark, NJ, USA), 100 g/mL penicillin, and 100 g/mL streptomycin. Cells were transfected with Lipofectamine 2000 (11668019, Invitrogen, Shanghai, China) according to the manufacturer’s protocol. Briefly, cells were trypsinized and seeded in 6-well plates at a rate of 1–3 million cells per well. The next day, 5 μg of RNLS overexpression plasmid or, a corresponding empty vector control plasmid (concentration estimated by O.D., plasmid synthesized by Genechem, Shanghai, China), was added to 250 μL of OPTI-MEM (Invitrogen) and mixed with an equal volume of OPTI-MEM containing 5 µL liposomes. The mixture was incubated at room temperature for 30 min and added directly to the cells.

### 2.9. Western Blot

The protein from NG, NTS, hearts from rats (4 rats per group), and PC12 cells were extracted using an RIPA buffer (Beyotime, Shanghai, China) complete protease inhibitor cocktail. PC12 cells were extracted with the RIPA buffer (Beyotime, Shanghai, China) complete protease inhibitor mixture. The protein concentration in the samples was measured by using the bicinchoninic acid protein assay kit (Beyotime, Shanghai, China). Each sample was separated on 5–15% SDS-PAGE and transferred to nitrocellulose membranes (Merck Millipore, MA, USA). The protein was further blocked with 5% non-fat dry milk for 1.5 h, and then incubated at 4 °C overnight with the primary antibodies for β-actin (WL01845, Wanleibio, Shenyang, China), Tyrosine Hydroxylase (ab129991), α-Syn (ab27766), RNLS (ab178700), and ALDH (ab215996), respectively. The densitometric analysis of the bands was performed using Odyssey 3.0 Software and all bands were normalized to β-actin levels as a control of the equal loading of samples in the total protein extracts.

Notably, the rat RNLS has 315 amino acid residues with a theoretical molecular mass of 34.95 kDa according to NCBI, and the antibody against RNLS used in this experiment is ~44 kDa, presumably due to the post-translational modification (e.g., glycosylation) or potentially due to splice variants, which can lead to a higher molecular mass on SDS-PAGE.

For the qRT-PCR, total RNA and cells were extracted from the rats (5 rats per group) using the TRIzol method. Total RNA (5 µg of total RNA) was reverse-transcribed using the High-Capacity cDNA Reverse Transcription Kit (Applied Biosystems, Waltham, MA, USA), and mRNA expression was determined using SYBR Green reagent (Applied Biosystems, Waltham, MA, USA) in an ABI 7500 Real-Time PCR System. The total volume of the PCR reaction system is 20 μL, containing 10 μL SYBR Green reagent, 0.4 μL forward, 0.4 μL reverse primer, and 1 μL cDNA template, and nuclease-free water was added to reach 20 μL. The PCR amplification program was set up for the following: Stage 1, holding at 95 °C for 10 min; Stage 2, cycling program at 95 °C for 15 s, 60 °C for 30 s, and then 72 °C for 30 s, for a total of 40 cycles; Stage 3,melting curve at 95 °C for 15 s, 60 °C for 1 min, and 60 °C to 95 °C for 15 s, with GAPDH as an internal control. The PCR primers were designed and synthesized by Invitrogen (Appendix A). Relative gene expression data were analyzed using the 2^−∆∆Ct^ method.

### 2.10. Immunohistochemical Analysis

PC12 cells were cultivated in 24-well plates. After fixing with 4% paraformaldehyde for 30 min, the cells were permeabilized with 0.4% Triton X-100 (Beyotime, Shanghai, China) for 30 min. Then, the cells were blocked with 50% bovine serum albumin (Sigma-Aldrich, St. Louis, MO, USA) for 1 h and incubated with antibodies (#ab129991, #ab27766, Abcam, Waltham, MA, USA) at 4 °C overnight. The next day, cells were thoroughly washed and incubated at room temperature for 1 h with an Alexa Fluor 488-conjugated antibody or Alexa Fluor 594-conjugated antibody. After washing, the nuclei were stained with DAPI (Beyotime, Shanghai, China).

### 2.11. High Performance Liquid Chromatography (HPLC)

After scraping the cells from the dish, the samples were placed into 10 mL EP tubes (4 mL cell suspension for each); 200 µL of chromatographic-grade methanol was added into the samples and mixed for 50 s. After that, 6 mL of acetone was added and mixed again for 120 s. Finally, the samples were centrifuged at 2000 rpm for 5 min. Once the liquid in the tube completely evaporated, the supernatant was transferred into a fresh 10 mL EP tube that was connected to a nitrogen tank to evaporate it at a consistent rate under nitrogen flow, and then frozen at 4 °C. Before injection, the sample/standard was reconstituted in methanol (chromatographically pure, 200 µL), and filtered through a microporous membrane; then, it was added to the injection bottle immediately (due to the strong reducibility of DOPAL), and automatically fed into the HPLC equipment (Agilent 1260 Infinity II, Agilent Technologies, Santa Clara, CA, USA) for data collection.

### 2.12. Transmission Electron Microscopy

The PC12 cells cultured in the six-well plate were collected and centrifuged into pellets, fixed in an electron microscopy solution, and then sliced using an ultramicrotome (UC-7, Leica, Beijing, China); the ultrathin sections were subjected to electronic staining and examined under an electron microscope (JEM-1220, JEOL, Ltd., Tokyo, Japan).

### 2.13. JC-1 and Calcein-AM/PI Double Staining

PC12 cells were cultured in 24-well plates, and the mitochondrial membrane potential and cell viability were detected using the Mitochondrial Membrane Potential Assay Kit (MA0338, Meilunbio, Dalian, China) and the CellTiter-Meiluncell Luminescent Cell Viability Assay Kit (PWL111-1, Meilunbio, Dalian, China).

### 2.14. CCK-8 Assay

The PC12 cell suspension was inoculated into 96-well plates with 200 µL per well. The plates were cultured in at 37 °C incubator for the required time. A total of 20 µL of CCK-8 solution (K1018, Cell Counting Kit-8, APEBIO, Shanghai, China) was added to each well and incubated for 2 h, and the absorbance was measured at 450 nm (SpectraMax M5, Molecular Devices, San Jose, CA, USA).

### 2.15. Statistical Analyses

All quantitative data are presented as mean ± standard deviation (SD). Statistical analyses were performed using Origin (Origin Lab Corporation, Northampton, MA, USA). Before applying parametric tests, data distribution was assessed for normality using the Shapiro–Wilk test. For comparisons between two groups, if data were normally distributed, a two-tailed unpaired Student’s *t*-test was used. For comparisons among multiple groups (three or more), if data were normally distributed and variances were found to be homogeneous, a one-way Analysis of Variance (ANOVA) was performed. If a significant difference was detected by the ANOVA, Bonferroni’s post hoc test was used for multiple comparisons to identify specific group differences. A *p* value < 0.05 was considered statistically significant. In figures, significance levels are denoted as *p* < 0.05, * *p* < 0.01, or ** *p* < 0.001, where applicable.

## 3. Results

### 3.1. Up-Regulated RNLS Expression in 6-OHDA-Treated Rats

There is strong evidence showing that early autonomic failure in patients with PD appears as overt orthostatic hypotension (OH) and/or supine hypertension (SH). At the 4th week post-6-OHDA injection, changes in blood pressure (BP) upon postural alterations (vertical vs. horizontal body position) were evaluated. The results showed that the BP declined when these model rats were placed at vertical position, with a substantially longer recovery time compared with the sham-control rats (Figure 1A,B). In stark contrast, BP was elevated while they were placed in a horizontal position, with a much longer recovery time compared with the sham group (Figure 1C,D), implying an impaired baroreflex function. Serum RNLS concentrations were subsequently measured in these model rats. In this regard, blood samples were taken from the 2nd to 5th week after 6-OHDA application. The results clearly indicated that RNLS began to rise markedly in the 3rd week and continued to rise gradually compared to the sham group (Figure 2A), which is at least one week earlier than the motor dysfunction manifested by behavioral tests/body rotation (Appendix A), strongly suggesting that the expression change in blood RNLS is very likely to be responsible for the abnormal metabolism of DA, and consequently impaired BP regulation through the baroreflex afferent function. Interestingly, by testing the model’s efficacy, we realized that not only systolic blood pressure (SBP), but also diastolic blood pressure (DBP) as well as pulse BP, were dramatically reduced at the 1st week after injection of 6-OHDA (Appendix A) without recovery at the 5th week, which is 3 weeks earlier than the significant change in the body rotation (Appendix A), indicating the RNLS-mediated direct impairment of cardiovascular activity.

To save time and experimental costs, the expression change in RNLS in the NG, NTS, and heart using these PD model rats were detected at the 4th week after 6-OHDA application, rather than the earlier weeks. The findings demonstrated that the expression of RNLS in the NG, NTS, and in the hearts of model rats was much higher than those of the sham group (Figure 2B–D). These data were strongly consistent with the notion that RNLS was dramatically increased at the 3rd week of 6-OHDA administration, and this change in RNLS occurs earlier than the symptoms of PD-like orthostatic BP and motor dysfunction observed prior to the 4th week after 6-OHDA injection.

### 3.2. Decreased DA Concentration in PC12 Cells in Parallel with RNLS Overexpression

As we explored the development of autonomic dysfunction in the PD model rats before motor dysfunction, we stumbled upon an increase in RNLS prior to the occurrence of autonomic dysfunction, which encouraged us to further explore the possible mechanisms of RNLS in the development of PD-like non-motor and behavioral changes. It is well recognized that DA is crucial to the pathological development of PD, and it has been reported that RNLS can regulate the level of DA in both central and peripheral organs [29]. For time and cost considerations, we selected rat adrenal pheochromocytoma PC12 cells to mimic dopaminergic neurons and explored the specific role of RNLS. The PCR results showed that the mRNA expression for RNLS was increased time-dependently after 24 h of transfection and slightly decreased at 72 h, but it remained at a higher level compared with the control group (Figure 3C). Immunoblotting results showed that at 48 h after transfection, the protein levels of RNLS were elevated (Figure 3A,B). Following RNLS overexpression in PC12 cells, we observed a statistically significant, albeit modest, time-dependent reduction in intracellular dopamine (DA) levels starting from 48 h post-transfection and persisting up to 96 h (Figure 3D). Specifically, at 48 h, DA levels decreased by approximately 6%, and this trend continued at 72 h (decrease of 12%) and 96 h (decrease of 18%) compared to control cells. These findings suggest that RNLS overexpression influences DA homeostasis in PC12 cells, leading to lower DA concentrations.

### 3.3. RNLS-Mediated DA Catalyzation with Increase in Its Metabolite Content

DOPAL is a toxic metabolite from the enzymatic deamination of DA. ALDH, a DOPAL detoxifying agent, may change it into harmless metabolites and expel it from cells [31]. Therefore, we detected the RNLS overexpression-mediated DOPAL level. The results showed that DOPAL was increased significantly at 48 h after the overexpression of RNLS, reached the highest level at 72 h, and then decreased slightly at 96 h, but remained at a higher level than the control group (Figure 4A,B). Western blot and PCR results showed that the expression of ALDH was also significantly up-regulated at 48 h after RNLS transfection (Figure 4C–E). These data highly imply that overexpressed RNLS is likely to metabolize DA directly and result in a reasonable increase in DOPAL, accompanied with the up-regulation of ALDH as a result, to compensate for the DOPAL increase.

### 3.4. Increased α-Syn Aggregation over the Time of RNLS Overexpression

α-Syn is a well-known pathological indicator of PD and plays an important role in the pathological process of PD. Studies have shown that DOPAL may be able to oligomerize α-Syn leading dopaminergic neurodegeneration [32]. In order to mimic this pathological change at the cellular level, we tested the expression of α-Syn monomers and multimers in PC12 cells overexpressing RNLS. The PCR results showed that the mRNA expression for α-Syn was significantly increased after 48 h overexpression (Figure 5A) and this result was in line with the immunofluorescent examination (Figure 5B,C). Western blot detection demonstrated that the α-Syn monomer was significantly increased at 48 h, and then slightly decreased at 72 h, with a higher level remaining compared with the control group, but was significantly lower than that of the control group at 96 h (Figure 5D,G). Dim-α-Syn was also significantly increased at 48 h, but was lower than that of the control group at 72 h through 96 h (Figure 5E,H). Tri-α-Syn was significantly increased after 48 h and 72 h and decreased significantly after 96 h and was lower than that of the control group (Figure 5F,I). These observations lead us to believe that α-Syn monomers could aggregate to dimers and consequently to trimers over time, leading to a gradual decrease in α-Syn monomers and dimers over time. Based on this observation, greater molecular weight aggregates were assumed to have declined tri-α-Syn levels at 96 h after treatment. To this end, the expression of α-Syn aggregates was further examined and the results showed that the protein band was formed more obviously at 170 kDa (Figure 6A,B) after the cells were treated for 96 h. Meanwhile, the immunohistochemical results showed an obvious fluorescence during the same time frame, and the total fluorescence intensity was greater than that of the control group (Figure 6C).

### 3.5. Reduced Activity of TH-Labeled Dopaminergic Neurons and PC12 Cells After RNLS overexpression

To further understand the impact of α-Syn on cell viability aggregation over time in response to DOPAL, the expression change in tyrosine hydroxylase (TH) was evaluated, since it is a rate-limiting enzyme for DA synthesis and is often used to label dopaminergic neurons [33]. After the cells were transfected for 48 h, the mRNA level of TH was reduced time-dependently (Figure 7A), with a similar trend and manner for the TH protein at 72 h after transfection (Figure 7B,C), which was also supported by the fact that the fluorescent intensity of TH-positive cells declined in the same time frame after transfection (Figure 7D,E). The next immediate question to be addressed was whether the function of these PC12 cells changed after transfection; so, viability was tested with the cell live and dead staining kit. Notably, the cell viability was decreased obviously at 72 h after transfection (Figure 8A,B), which was also confirmed with CCK-8 detection (Figure 8C).

### 3.6. Impaired Axonal Transport over Time in PC12 Cells Followed by Transfection

Impaired axonal transportation exists in PD and a study has shown that α-Syn mutation and overexpression may lead to axonal transport defects [34], which is presumably a key contributor to neuronal degeneration in the baroreflex afferent pathway. Therefore, the expression of axonal transportation-related proteins such as kinesin and dynein was detected over time. PCR results showed that the mRNA for dynein was down-regulated time-dependently in PC12 cells 48 h after transfection (Figure 9A), with a similar trend and manner for dynein protein (Figure 9B,C). For kinesin, mRNA (Figure 9D) and proteins (Figure 9E,F) were both down-regulated time-dependently starting at 48 h after transfection, highly suggesting that α-Syn aggregation also injures axonal transportation by down-regulating axon transport-related proteins, taking place robustly much earlier than α-Syn aggregation-mediated neuronal degeneration.

### 3.7. Ultrastructure Changes in PC12 Cell After Transfection

Abnormal organelle function is common in PD patients and is closely associated with α-Syn aggregation [35]. Changes in mitochondrial membrane potential, an indicator of mitochondrial function, were assessed using a JC-1 kit in PC12 cells following RNLS overexpression. The results showed that the mitochondrial membrane potential was significantly decreased at 72 h after transfection (Figure 10A,B), which confirmed by ultrastructural changes observed by electron microscopy, indicating clear early apoptosis, nuclear pyknosis, mitochondria swelling, and cytoplasm swelling. The number of early apoptosis was increased at 96 h, alongside aggravated mitochondrial injury and cytoplasmic swelling, implying that α-Syn begins to aggregate at 48 h and is followed by neuronal sub-organelles being damaged at 72 h after RNLS transfection; these changes cause expected morphological pyknosis, swelling, and apoptosis over time at the cellular level (Figure 10C).

## 4. Discussion

Currently, no effective treatment is available for PD, a neurodegenerative disorder marked by the gradual loss of dopaminergic neurons in the substantia nigra and the abnormal accumulation of Lewy bodies, leading to the characteristic motor manifestations of the disease. The prodromal phase, often presenting with non-motor symptoms, may occur more than 20 years before the onset of motor impairment [19]. In fact, when motor symptoms first appear at the outset of PD, patients frequently have already lost 60% of the dopaminergic neurons from the SNpc [36]. One of the most representative investigations has demonstrated that α-Syn prefabricated fibers could be injected into rats from the peripheral site and diffused into the center location, and this process-mediated diffusion would have different pathological changes upon the location of α-Syn [37,38]. As a result, more research has been focusing on the peripheral etiology of PD. More importantly, the insights for the peripheral pathophysiology of α-Syn aggregation remain largely unknown, which motivated the current investigation.

Based upon the gut–brain axial hypothesis, it was hypothesized that α-Syn misfolded aggregation is induced by inflammation via microbial pathogens and propagates through the gastrointestinal tract, and ultimately leads to the death of dopaminergic neurons [39,40,41]. In our initial in vivo study, elevated levels of RNLS were detected in a 6-OHDA-treated PD model rat with predated orthostatic BP (Figure 1), which sparked our interest in exploring whether there is a link between RNLS and PD development. Intriguingly, several lines of clinical evidence support our research interest, including the following: (1) higher DA levels in schizophrenia are associated with reduced RNLS levels [42]; (2) RNLS was initially increased with BP reduction in renal denervated rats and followed by a decrease in RNLS and increased BP [43]. In the current study, differentiated PC12 cells were selected, which perfectly fits our ideal cellular model to be capable of forming functional synapses and secreting DA [44,45]. In our PC12 cell model, the overexpression of RNLS led to a statistically significant reduction in DA levels, although the magnitude of this change was modest (Figure 3D). It is important to consider that even subtle, sustained alterations in DA homeostasis can have significant downstream consequences, particularly in the context of chronic neurodegenerative processes. The accumulation of toxic metabolites like DOPAL, even if resulting from a small but persistent shift in DA metabolism, could progressively contribute to α-Synuclein aggregation and neuronal stress over extended periods. Furthermore, PC12 cells, while a valuable model for dopaminergic neurons, may possess robust compensatory mechanisms or differ in their metabolic flux compared to primary neurons in vivo, potentially attenuating the observed decrease in DA. Nevertheless, the consistent reduction in DA, coupled with the subsequent increase in DOPAL (Figure 4A,B) and α-Syn aggregation (Figure 5 and Figure 6), underscores a detrimental role for RNLS overexpression in pathways relevant to PD pathology. These findings indicate that RNLS overexpression influences DA metabolic pathways, leading to increased DOPAL formation in these cells, potentially by altering the activity of enzymes directly involved in DA metabolism or by affecting the cellular environment in a way that favors DOPAL production. We also detected a decreased level of TH as α-Syn started to increase, whereas the level of the TH protein declined, along with reduction in cell viability at 72 h throughout the process of α-Syn aggregation (Figure 7 and Figure 8), which might be another plausible reason for the decrease in DA levels in addition to the RNLS, since TH is the rate-limiting enzyme necessary for DA production [46]. We also observed the down-regulation of axonal transport-related proteins and mitochondrial damage during α-Syn aggregation (Figure 9 and Figure 10).

By combining the literature reports as supporting evidence with our experimental results, a summary was made to visually display the outline of the research background, hypothesis, and focus of this study (Figure 11). Taken together, the current investigation develops a new cellular model by overexpressing RNLS in PC12 cells, demonstrating that this overexpression leads to alterations in DA metabolism (specifically, reduced DA and increased DOPAL), which subsequently induce α-Syn aggregation. Its significance is to better simulate the peripheral pathogenesis of PD at the cellular level. α-Syn also accumulates in other synucleinopathies, including dementia with Lewy bodies (DLB), multiple system atrophy (MSA), and various lysosomal storage disorders. Importantly, α-Syn regulates the fibrosis of Aβ and tau, two key proteins in the pathophysiology of Alzheimer’s disease (AD) [47]. Therefore, exploring the source of α-Syn is crucial for evaluating potential treatments for synucleinopathies. This target may prove to be of general utility and provide a basis in future treatments for most forms of PD, as well as potentially other synaptic nucleopathies. Compared with the currently and widely used 6-OHDA or MPTP modeling, it can better characterize chronic pathological changes in DA neurons, and explore the possible mechanism of the accumulation of α-Syn pathological markers instead of killing neurons quickly; this also suggests that RNLS may be used as one of the biomarkers for the early onset of PD. But on the other hand, we still do not know whether the process of neuronal damage is caused by the damage of the axonal transport or organelle dysfunction caused by the aggregation of α-Syn. We still have not reached a clear conclusion about the sequence of this series of damage processes. PD is a complicated and ever-changing disease and is accompanied by numerous clinical challenges, including timely diagnosis in the early stages.

Our study is not without limitations. A clear causal link between RNLS and early autonomic disturbances could not be clearly established due to the lack of direct evidence of RNLS overexpression matching the time point of BP changes, although RNLS-mediated BP disturbance/non-motor symptoms did appear before body rotation/motor impairment. Consequently, the aforementioned causal link remains hypothetical at this stage, necessitating further investigation with more temporally focused in vivo studies.

## 5. Conclusions

Our study provides compelling evidence that the overexpression of RNLS plays a significant role in initiating key pathological events relevant to the peripheral etiology and early stages of PD. We demonstrate that elevated RNLS levels lead to a dysregulated DA metabolism, characterized by reduced DA and an increase in its highly toxic metabolite, DOPAL. This accumulation of DOPAL, driven by RNLS overexpression, subsequently promotes the time-dependent aggregation ofα-Syn, a hallmark pathological protein in PD, within cellular models mimicking baroreflex afferent neurons.

Furthermore, our findings link these molecular events to functional impairments. The RNLS-induced α-Syn aggregation was associated with impaired axonal transport mechanisms and culminated in neuronal damage, as evidenced by ultrastructural changes and reduced cell viability. In our in vivo PD model, increased RNLS expression was correlated with early signs of autonomic dysfunction, specifically orthostatic blood pressure changes, which manifested prior to significant motor deficits. This underscores the potential contribution of RNLS-mediated pathology in baroreflex afferents to the early non-motor symptoms of PD.

Collectively, these observations highlight a detrimental cascade triggered by RNLS overexpression: excessive peripheral DA metabolism, DOPAL toxicity, α-Syn aggregation, axonal transport defects, and ultimately, neuronal degeneration within the baroreflex afferent pathway. These insights not only reinforce the concept of a peripheral origin for PD but also identify RNLS as a potential early biomarker and a novel therapeutic target. Future investigations aimed at understanding the precise upstream regulators of RNLS expression and further delineating the sequence of these pathogenic events could pave the way for new strategies to diagnose and intervene in the early phases of PD and related synucleinopathies.

## Figures and Tables

**Figure 1 biomedicines-13-01243-f001:**
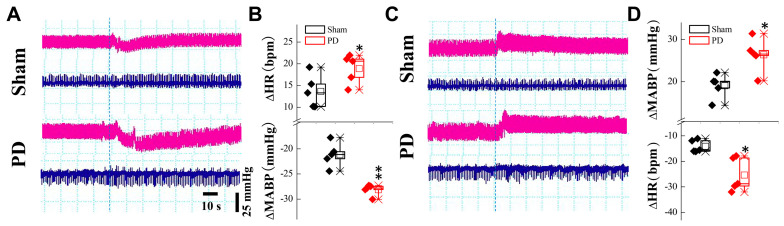
BP changes while body position changed in DOPAL 6-OHDA-treated PD model rats. (**A**–**D**) The representative changes in BP (pink traces) and heart rate (HR, blue traces) at orthostatic or supine status compared with steady state; differences were expressed as mean ± SD. * *p* < 0.05 and ** *p* < 0.01 vs. sham, *n* = 5 rats per group.

**Figure 2 biomedicines-13-01243-f002:**
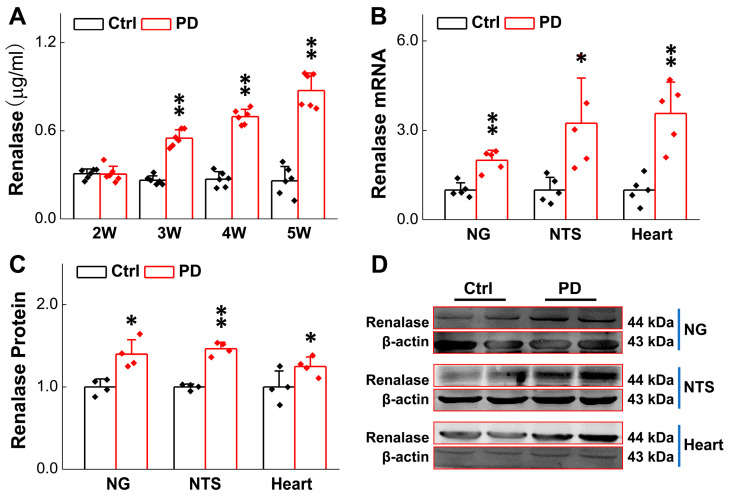
Expression of renalase in rats was increased after 6-OHDA treatment. (**A**) Detection of serum renalase concentration from the 2nd to 5th week after 6-OHDA application by ELISA kit (*n* = 6 rats per group). (**B**) qRT-PCR detection of relative renalase expression in NG, NTS, and heart tissues of Parkinson’s rats and sham group (4th week after 6-OHDA application, *n* = 5 rats per group). (**C**,**D**) Western blot assay showing relative protein levels of renalase in NG, NTS, and heart tissues of Parkinson’s rats and sham group (*n* = 4 rats per group). Data are presented as mean ± SD, * *p* < 0.05 and ** *p* < 0.01 vs. sham.

**Figure 3 biomedicines-13-01243-f003:**
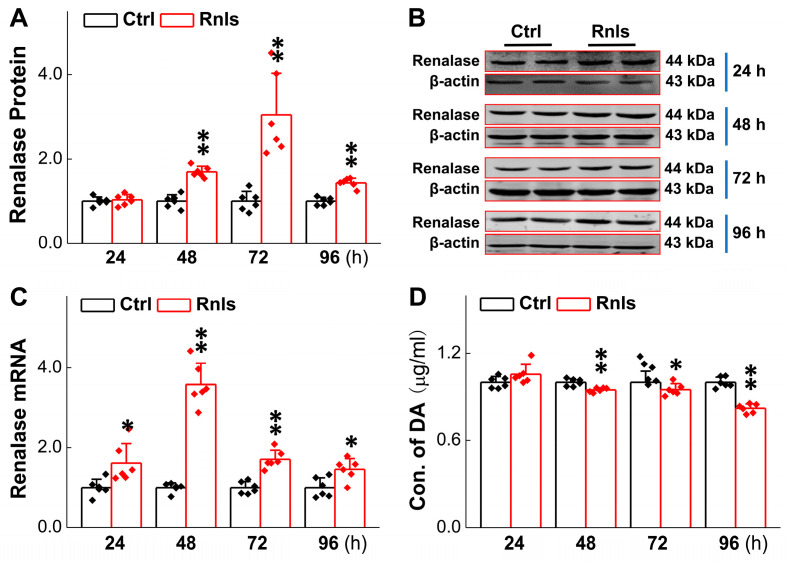
PC12 cells transfected with RNLS overexpression plasmid (or empty vector control) continued to overexpress and metabolize DA. (**A**,**B**) Transient transfection of PC12 cells with RNLS overexpression plasmid (or empty vector control) expressing the protein. Western blot assay showing relative protein levels of RNLS in PC12 cells at different time points (*n* = 6). (**C**) qRT-PCR detection of relative RNLS at different time points after PC12 cells transfected with (*n* = 6). (**D**) Detection of relative dopamine (DA) concentration in PC12 cells at different time points by ELISA kit following RNLS overexpression or NC, showing altered DA levels (*n* = 6). Data are presented as mean ± SD, * *p* < 0.05 and ** *p* < 0.01 vs. Ctrl.

**Figure 4 biomedicines-13-01243-f004:**
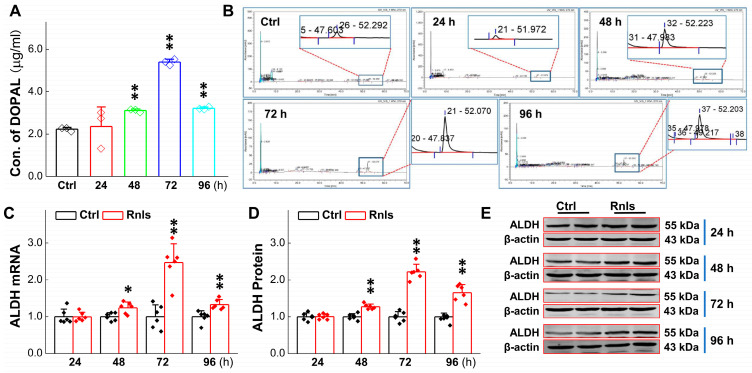
Increased levels of DA metabolites after RNLS overexpression of in PC12 cells. (**A**,**B**) Detection of DOPAL content at different time points in PC12 cells after RNLS overexpression by high performance liquid chromatography (*n* = 3). (**C**) qRT-PCR detection of relative ALDH expression at different time points after PC12 cells were transfected with (*n* = 6). (**D**,**E**) Western blot analysis showing relative protein levels of ALDH at different time points after PC12 cells were transfected with (*n* = 6). Data are presented as mean ± SD, * *p*< 0.05 and ** *p* < 0.01 vs. Ctrl.

**Figure 5 biomedicines-13-01243-f005:**
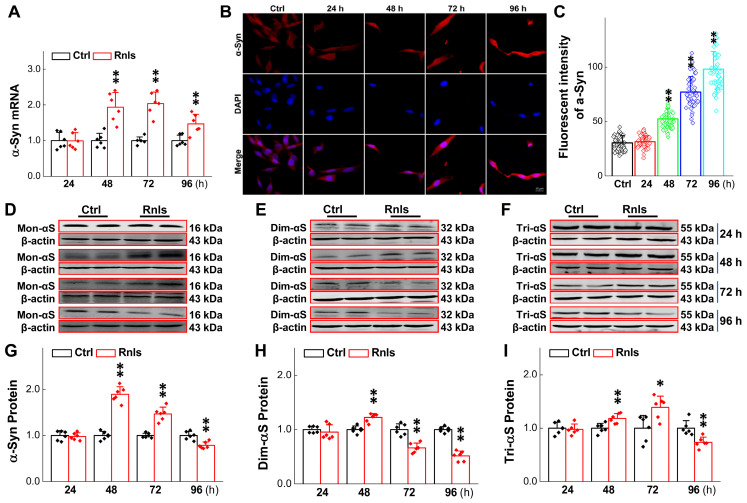
Changes in α-Syn content at different time points in RNLS-overexpressing PC12 cells. (**A**) qRT-PCR detection of relative α-Syn expression at different time points after PC12 cells transfected with RNLS (*n* = 6). (**B**) Representative image of immunofluorescence with enhanced α-Syn fluorescence after transfection of PC12 cells. Scale bar: 20 μm. (**C**) Immunohistochemical quantitative statistics (*n* = 60). (**D**–**G**) Western blot analysis showing relative protein levels of monomer of α-Syn (Mon-αS) at different time points after PC12 cells were transfected with RNLS (*n* = 6). (**E**–**H**) Western blot analysis showing relative protein levels of dimer of α-Syn (Dim-αS) at different time points after PC12 cells were transfected with RNLS (*n* = 6). (**F**–**I**) Western blot analysis showing relative protein levels of trimer of α-Syn (Tri-αS) at different time points after PC12 cells were transfected with RNLS (*n* = 6). Data are presented as mean ± SD, * *p* < 0.05 and ** *p* < 0.01 vs. Ctrl.

**Figure 6 biomedicines-13-01243-f006:**
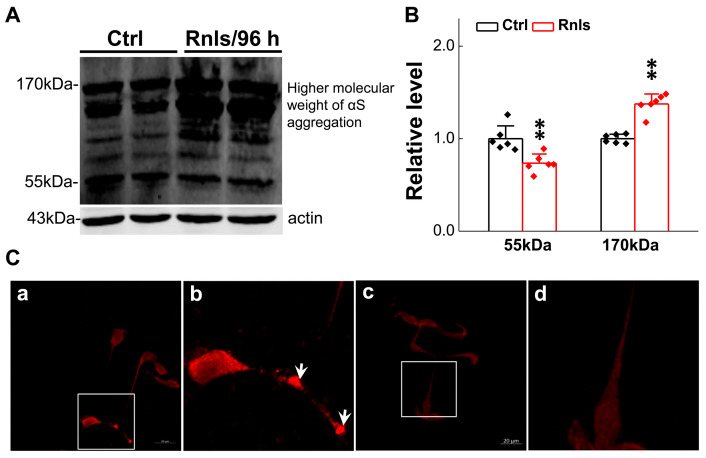
α-Syn aggregated in PC12 cells transfected with RNLS for 96 h. (**A**,**B**) Representative graph of Tri-αS and higher molecular weight α-Syn aggregates in PC12 cells transfected with RNLS for 96 h (*n* = 6). (**C**(**a**,**b**)) Immunofluorescent photomicrographs of transfected cells (overexpressing RNLS protein for 96 h), cytoplasmic aggregates in the cell neurotic processes are highlighted by white arrows in the figure. (**C**(**c**,**d**)) Untreated control PC12 cells. Scale bar: 20 μm. Data are presented as mean ± SD, ** *p* < 0.01 vs. Ctrl.

**Figure 7 biomedicines-13-01243-f007:**
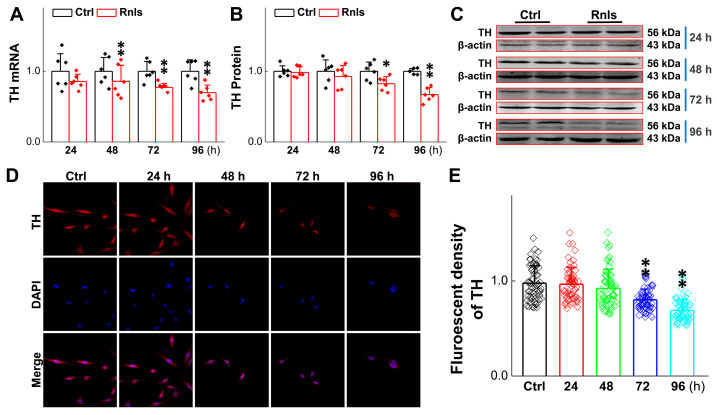
The content of tyrosine hydrolase (TH) decreased after PC12 cells were transfected with RNLS. (**A**) qRT-PCR detection of relative TH expression at different time points after PC12 cells transfected with RNLS (*n* = 6). (**B**,**C**) Western blot analysis showing relative protein levels of TH at different time points after PC12 cells were transfected withR NLS (*n* = 6). (**D**) Representative graph of immunofluorescent (IF) results of TH in PC12 cells at different time points administered with RNLS overexpression. Scale bar: 20 µm. (**E**) Immunohistochemical quantitative statistics (*n* = 60). Data are presented as mean ± SD, * *p* < 0.05 and ** *p* < 0.01 vs. Ctrl.

**Figure 8 biomedicines-13-01243-f008:**
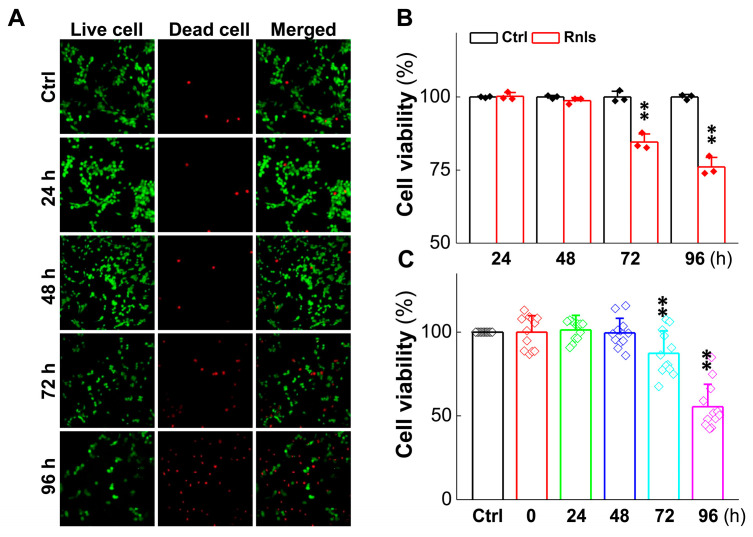
Cell viability decreased after PC12 cells were transfected with RNLS. (**A**) Live/dead cell viability assay of cultured PC12 cells. The cells were transfected with plasmids and cultured for 24, 48, 72, or 96 h and then stained with the Calcein-AM/PI Double Staining Kit. The live and dead cells exhibited green and red fluorescence. Scale bar: 10 µm. (**B**) Immunohistochemical quantitative statistics (*n* = 3). (**C**) CCK-8 was used to determine PC12 cells viability following overexpression or NC for 24, 48, 72 or 96 h (*n* = 12). Data are presented as mean ± SD, ** *p* < 0.01 vs. Ctrl.

**Figure 9 biomedicines-13-01243-f009:**
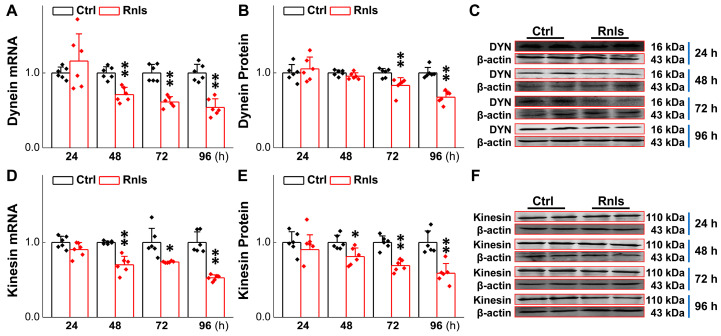
Decreased expression of axonal transporters after RNLS overexpression in PC12 cells. (**A**) qRT-PCR detection of relative dynein (DYN) expression at different time points after PC12 cells transfected with RNLS (*n* = 6). (**B**,**C**) Western blot analysis showing relative protein levels of dynein at different time points after PC12 cells were transfected with RNLS (*n* = 6). (**D**) mRNA expression of relative kinesin in PC12 cells at different time points (*n* = 6). (**E**,**F**) Western blot analysis showing relative protein levels of KHC at different time points after PC12 cells were transfected with RNLS (*n* = 6). Data are presented as mean ± SD, * *p* < 0.05 and ** *p* < 0.01 vs. Ctrl.

**Figure 10 biomedicines-13-01243-f010:**
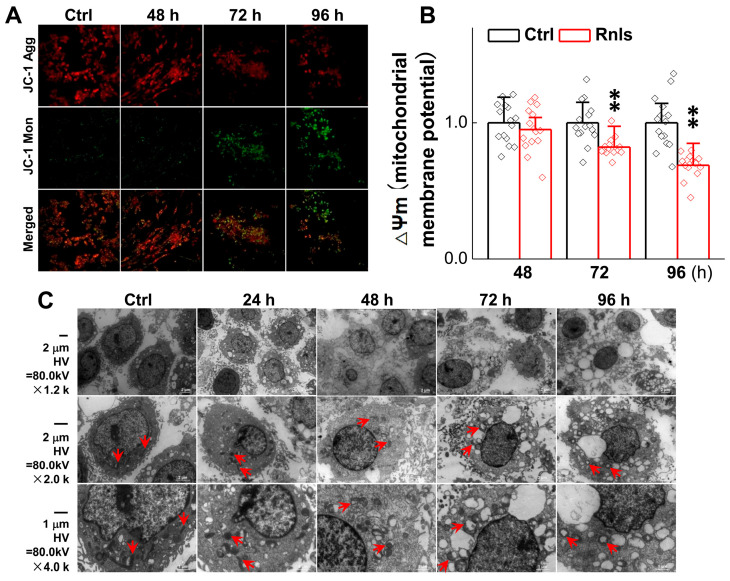
Ultrastructural damage after RNLS overexpression in PC12 cells. (**A**,**B**) The relative ∆Ψ m in PC12 cells for each treatment at different time points, as determined by JC-1 staining (*n* = 15). Data are presented as mean ± SD, ** *p* < 0.01 vs. Ctrl; scale bar: 20 µm; ∆Ψm: mitochondrial membrane potential. (**C**) Representative images of transmission microscopy at different time points after PC12 cells were transfected with RNLS. Scale bar: 1 µm and 2 µm, direct magnification: ×1200, ×2000 and ×4000, the red arrow indicates mitochondria.

**Figure 11 biomedicines-13-01243-f011:**
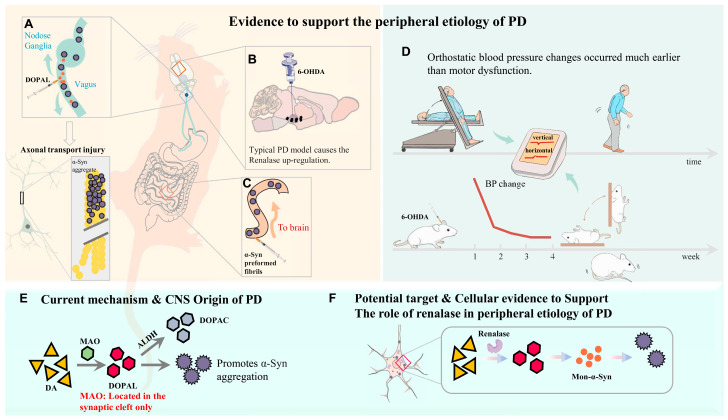
Numerous clues to PD peripheral pathology. (**A**) Vagal application of DOPAL (3,4-dihydroxyphenylacetaldehyde) to simulate PD-like autonomic dysfunction strongly suggests that DOPAL will likely induce the expression and aggregation of α-Syn/oligomers at the injection site, which will then be transported to the heart and the NG. Axonal transportation is likely to be impaired by DOPAL and DOPAL-mediated oligomer α-Syn [22]. (**B**) Typical PD model causes the RNLS up-regulation. (**C**) The pathological α-Syn protein fibers were injected into the mouse intestine, and it was found that the α-Syn protein eventually diffused into the substantia nigra pars compacta, where it degraded dopaminergic neurons. (**D**) Orthostatic hypotension (OH) has been linked to a higher risk of Parkinson’s disease in studies [48,49]. OH is a predictor of motor decline in individuals with early PD [50]. The experiment confirmed that the change in BP also occurred before the movement disorder in rats treated with 6-OHDA. (**E**) Current mechanisms and central nervous system origins of PD: MAO-mediated DA metabolism in the central nervous system. (**F**) The role of RNLS in peripheral etiology of PD.

## Data Availability

The original contributions presented in the study are included in the article/Appendix A, further inquiries can be directed to the corresponding author.

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
