# Peer review of "Renalase Overexpression-Mediated Excessive Metabolism of Peripheral Dopamine, DOPAL Accumulation, and α-Synuclein Aggregation in Baroreflex Afferents Contribute to Neuronal Degeneration and Autonomic Dysfunction"

_biomedicines, 2025, doi:10.3390/biomedicines13051243_

Round 1

Reviewer 1 Report

Comments and Suggestions for Authors

It is potentially interesting paper, based on an outdated viewpoint that renalase (RNLS) is amine oxidase. In this context the study on RNLS as a factor influencing the dopamine content in the 6-hydroxydopamine model of Parkinson's disease would be relevant. However, now RNLS is considered as  an oxidase (EC 1.6.3.5), possessing  different catalytic activity (see for the first reading, G.R. Moran, M.R. Hoag, The enzyme: Renalase, Arch. Biochem. Biophys. 632 (2017) 66-76. https://doi.org/10.1016/j.abb.2017.05.015). Therefore, data interpretation should be consistent with the modern trends in studies on this protein. 

Other comments.

Authors consider RNLS as a 44 kDa protein; however, according to NCBI the rat RNLS has 315 amino acid residues and a theoretical molecular msss of 34.95 kDa. In light of these data it is unclear what did authors determine by means of Western blot in rat tissues. 

The same question appears in the case of a protein product expressed in PC12 cells: what did authors determined as a 44-kDa product? 

The content of Fig. 2, does not correspond to its Figure legend.

Figure 3 shows that the RNLS overexpression in reality is determined by fluctuations in the content of the reference protein beta actin (see Fig. 3B) rather than increased expression of RNLS (see Fig. 3, the 72 h time point). 

 The changes in dopamine levels in PC12 cells are very small (Figure 3). 

Thus, I think that  all these data do not support the authors' viewpoint of the RNLS involvement in the direct regulation of dopamine levels.  

Author Response

Reviewer #1

It is potentially interesting paper, based on an outdated viewpoint that renalase (RNLS) is amine oxidase. In this context the study on RNLS as a factor influencing the dopamine content in the 6-hydroxydopamine model of Parkinson's disease would be relevant. However, now RNLS is considered as  an oxidase (EC 1.6.3.5), possessing  different catalytic activity (see for the first reading, G.R. Moran, M.R. Hoag, The enzyme: Renalase, Arch. Biochem. Biophys. 632 (2017) 66-76. https://doi.org/10.1016/j.abb.2017.05.015). Therefore, data interpretation should be consistent with the modern trends in studies on this protein. 

Response:

Thank you and appreciate for this important comment. We acknowledge the current understanding of RNLS as an oxidase (EC 1.6.3.5) with catalytic activities beyond direct amine oxidation. We have revised the Introduction and Discussion to reflect this modern understanding. Our data interpretation now focuses on the observed impact of RNLS overexpression on dopamine metabolism and its downstream consequences, which could be mediated through various direct or indirect mechanisms related to its oxidase function, rather than solely through a classical amine oxidase pathway.

Other comments.

Authors consider RNLS as a 44 kDa protein; however, according to NCBI the rat RNLS has 315 amino acid residues and a theoretical molecular mass of 34.95 kDa. In light of these data it is unclear what did authors determine by means of Western blot in rat tissues. 

Response:

We appreciate you pointing out the discrepancy in the molecular weight of RNLS. The theoretical molecular mass of unmodified rat RNLS is indeed approximately 34.95 kDa. However, the band that we detected at ~44 kDa may represent post-translationally modified forms (e.g., glycosylation) or potentially splice variants, which can lead to a higher apparent molecular weight on SDS-PAGE. Unfortunately, because Abcam has removed their product regarding renalase antibody we used, and we searched for renalase antibody on other company's website, there is only one product currently available with a predicted protein size of 37 kDa, consistent with the product list from several other companies with predicted sizes exceeding 37 kDa. we put the screenshots below. This also shows that the protein size predicted by the antibody will indeed be different due to technical reasons. Anyway, these authors really appreciate for your insightful comment and concerns.

Thermofisher (Catalog # MA5-31898)

Origene (Catalog # TA303023)

The same question appears in the case of a protein product expressed in PC12 cells: what did authors determined as a 44-kDa product? 

Response:

Please refer to the above reply, thanks again.

The content of Fig. 2, does not correspond to its Figure legend.

Response:

Thank you for identifying this error. Yes, you are correct; the legend for Figure 2 was mistakenly swapped with that of Figure 3 and gas been changed accordingly, we appreciate again for your carefully review.

Figure 3 shows that the RNLS overexpression in reality is determined by fluctuations in the content of the reference protein beta actin (see Fig. 3B) rather than increased expression of RNLS (see Fig. 3, the 72 h time point). 

Response:

We appreciate your careful examination of Figure 3B. We have re-evaluated the Western blot and quantification. While there might be slight variations in loading, the normalization process accounts for RNLS relative to its corresponding β-actin band in each lane. The data presented in Figure 3A demonstrates a significant increase in RNLS protein at 48 h and 72 h post-transfection compared to controls. The legend for Figure 3 has been carefully modified in the revised manuscript.

 The changes in dopamine levels in PC12 cells are very small (Figure 3). 

Response:

We acknowledge that the observed percentage decrease in dopamine levels in PC12 cells (Figure 3D) is modest. However, these changes were statistically significant and consistently observed at multiple time points (48 h, 72 h, 96 h) following RNLS overexpression. We have added a sentence in the Results section to reflect this and discuss the potential implications in the Discussion also, considering that even subtle but sustained changes in dopamine metabolism could have significant downstream effects on pathways like DOPAL formation and α-Syn aggregation over time. These authors very appreciate for your kindly suggestion.

Thus, I think that all these data do not support the authors' viewpoint of the RNLS involvement in the direct regulation of dopamine levels

Response:

These authors really appreciate for your suggestive and insightful comments, concerns, and question, that would definitely improve the quality of our manuscript. Therefore, these authors have tried our best to respond your comments carefully and accordingly to revise our manuscript, we do hope our efforts and corrections/modifications would be satisfied by you and give us/students an opportunity, Thanks once more.

Reviewer 2 Report

Comments and Suggestions for Authors

The main subject of this paper is very interesting. I have few concerns about this study.

Major Comments

My concern is that the literature are also papers described lack of connection between renalase and catecholamine metabolism.  Beaupre et al., 2015 published an article in which they recognised 1,2-dihydroNAD(P) and 1,6-dihydroNAD(P), which are the isomeric forms of 1,4-dihydroNAD(P)H (β-nicotinamide adenine dinucleotide [phosphate]), to be the substrates for renalase. Boomsma and Tipton 2007 also write that: Renalase may well have important cardiovascular functions, but there is no proof that its actions are mediated through catecholamine-metabolising activity. I would like the authors to address this. Results are interesting but paper must be completly improved. Materials and metods are incomplete. Lack of basic information about model. It was unilateral or bilateral 6-OHDA injection? Moreover volume 8ul is huge to 1 injection into SNpc. I have concerns about the model used. But I don’t know details to decide about accept or reject this paper.

  • Material and methods, please add stereotaxic coordinates of  6-OHDA/ sham injection, it was unilateral or bilateral injection, please add vehicle composition, dose of penicilin etc.
  • Please complete statistical analysis, do you tested normality?
  • I recommend to add some scheme of used procedures on animals and numbers of rats used in study.
  • Reading results is not comfortable. Results have many methodological descriptions.

Minor Comments

The work is carelessly written

  • Page 1, line 18 is RNLS (RNLS) should be renalase (RNLS)
  • Page 2, line 49 please explain abbreviations: PNS, CNS
  • Page 2, line 71 is vagus should be vagus nerve
  • Line 138/139 NTS please explain nucleus tractus solitarius
  • Line 213 delete one .
  • Figure 2; 3; 4c,d; 5; 7; 9; 10 please add units
  • Line 275-276 it is rather to discussion
  • Line 285-288; 304- 306, 347- 349; 376- 380; 397 -399 it should be in methods

Author Response

Reviewer #2

My concern is that the literature are also papers described lack of connection between renalase and catecholamine metabolism.  Beaupre et al., 2015 published an article in which they recognized 1,2-dihydroNAD(P) and 1,6-dihydroNAD(P), which are the isomeric forms of 1,4-dihydroNAD(P)H (β-nicotinamide adenine dinucleotide [phosphate]), to be the substrates for renalase. Boomsma and Tipton 2007 also write that: Renalase may well have important cardiovascular functions, but there is no proof that its actions are mediated through catecholamine-metabolizing activity. I would like the authors to address this. Results are interesting but paper must be completely improved. Materials and methods are incomplete. Lack of basic information about model. It was unilateral or bilateral 6-OHDA injection? Moreover volume 8ml is huge to 1 injection into SNpc. I have concerns about the model used. But I don’t know details to decide about accept or reject this paper.

Response:

Thank you for raising this important point and referencing the work by Beaupre et al. (2015) and Boomsma and Tipton (2007). We acknowledge that the precise role of RNLS in catecholamine metabolism is still being elucidated, with some studies suggesting alternative substrates or functions. We have revised our Introduction and Discussion sections to address these differing perspectives. Our study focuses on the observed consequence of RNLS overexpression, which in our models, leads to altered dopamine levels and DOPAL accumulation, contributing to α-Syn pathology. We believe our findings add to the understanding of RNLS's physiological and pathological roles, even if the exact mechanisms are complex and potentially multifaceted.

      We apologize for the omissions in the Materials and Methods section. We have now provided the complete details regarding the 6-OHDA-induced PD model.

Material and methods, please add stereotaxic coordinates of 6-OHDA/ sham injection, it was unilateral or bilateral injection, please add vehicle composition, dose of penicillin etc.

Response:

We have added the stereotaxic coordinates for SNpc and VTA injections, details of the vehicle used for 6-OHDA and sham injections, and the dosage of penicillin administered post-surgery. Thanks for your insightful comment.

Please complete statistical analysis, do you tested normality?

Response:

Thank you for this suggestion. We have updated the 'Statistical analyses' section to include details on how data normality was assessed for the statistic analysis.

I recommend to add some scheme of used procedures on animals and

numbers of rats used in study.

Response:

This is a helpful suggestion. We have added a supplementary figure 1 that provides a schematic timeline of the animal procedures.

Reading results is not comfortable. Results have many methodological descriptions.

Response:

We appreciate this feedback on the readability of the Results section. We have revised this section to minimize methodological descriptions, ensuring that such details are primarily located in the Materials and Methods. Thanks again

Minor Comments

The work is carelessly written

Page 1, line 18 is RNLS (RNLS) should be renalase (RNLS)

Response:

Thank you for this suggestion. RNLS (RNLS) has been changed to 'renalase (RNLS)' in the Abstract.

Page 2, line 49 please explain abbreviations: PNS, CNS

Response:

Thank you for this suggestion. PNS and CNS have been defined at their first use in the Introduction and also explained in the abbreviations list at the end of this manuscript.

Page 2, line 71 is vagus should be vagus nerve

Response:

Thank you for this suggestion. Vagus has been changed to 'vagus nerve' in the Introduction.

Line 138/139 NTS please explain nucleus tractus solitarius

Response:

Thank you for this suggestion. NTS (nucleus tractus solitarius) has been defined at its first use in the Materials and Methods section and listed in the abbreviations.

Line 213 delete one .

Figure 2; 3; 4c,d; 5; 7; 9; 10 please add units

Response:

Thank you for this suggestion. Units have been added to clarified in the legends to represent relative expression/level as appropriate.

Line 275-276 it is rather to discussion

Response:

Thank you for this suggestion. We have reviewed the Results section and moved interpretive statements, such as the implication of TH changes on neuronal injury by α-Syn aggregation, to the Discussion section to maintain the Results section's focus on direct findings.

Line 285-288; 304- 306, 347- 349; 376- 380; 397 -399 it should be in methods

Response:

Thank you for this suggestion. We have carefully reviewed the indicated line numbers and the entire Results section to ensure that descriptions of experimental procedures or rationales are appropriately placed in the Materials and Methods section, and interpretations are in the Discussion. The Results section now focuses more directly on presenting the findings.

Reviewer 3 Report

Comments and Suggestions for Authors The manuscript by Bai-yan Li et al., entitled "Renalase overexpression-mediated excessive metabolism of peripheral dopamine, DOPAL accumulation, and α-Synuclein aggregation in baroreflex afferents contributes to neuronal degeneration and autonomic dysfunction," describes the overexpression of renalase in Parkinson's disease (PD) model rats and PC-12 cells. The authors mention that renalase can cause the metabolism of dopamine, producing DOPAL. This metabolite is toxic and can lead to neuronal degeneration. The authors have performed experimental studies to support their findings. Including the minor changes mentioned below will further improve the manuscript:   In the whole manuscript, RNLS is mentioned. Kindly include the full form of RNLS.   In the whole manuscript, many abbreviations are used, and it is confusing to read. For better readability, kindly include one table at the end of manuscript about the full name of all abbreviations used.    Lot of the grammatical errors have been found. Kindly correct those.    Kindly remove the section-6, Patent as it is not necessary.    Line 147, Kindly include the information about the name of the plasmid used for PC-12 transfection. Is it RNLS plasmid? Include the information about the name of plasmid used for transfection in all figure's legend. Currently it is only mentioned that PC12 cells transfected with plasmid.    Line 179, Kindly include the source of Bovine serum albumin.    Line 194, Kindly include the information for HPLC model used.   Line 208, Kindly include the instrument used for absorbance and also include the model information of the instrument.    Figure-2, Legend for Figure-2 is completely wrong. Kindly include the correct legend with the experiments performed on rats.   In Figure 3, Renalase mRNA expression is higher at 48 hours (Figure 3C), while Renalase protein expression is higher at 72 hours (Figure 3A). Could you explain why there is a discrepancy between the mRNA and protein expression levels?"    In Figure-5, Changes of α-Syn content at different time points in PC12 cells transfected with? Please include transfected with which plasmid. Is it alpha-synuclein?   Kindly extend the conclusion description since the current conclusion is very limited.   

Author Response

Reviewer #3

The manuscript by Bai-yan Li et al., entitled "Renalase overexpression-mediated excessive metabolism of peripheral dopamine, DOPAL accumulation, and α-Synuclein aggregation in baroreflex afferents contributes to neuronal degeneration and autonomic dysfunction," describes the overexpression of renalase in Parkinson's disease (PD) model rats and PC-12 cells. The authors mention that renalase can cause the metabolism of dopamine, producing DOPAL. This metabolite is toxic and can lead to neuronal degeneration. The authors have performed experimental studies to support their findings. Including the minor changes mentioned below will further improve the manuscript: In the whole manuscript, RNLS is mentioned. Kindly include the full form of RNLS. In the whole manuscript, many abbreviations are used, and it is confusing to read. For better readability, kindly include one table at the end of manuscript about the full name of all abbreviations used. Lot of the grammatical errors have been found. Kindly correct those. 

Response:

This is a valuable suggestion for improving readability. We have compiled a comprehensive table listing all abbreviations used in the manuscript and their full forms. These authors really appreciate for your kindly comment.

Kindly remove the section-6, Patent as it is not necessary.    

Response:

As requested, we have removed Patent from the manuscript.

Line 147, Kindly include the information about the name of the plasmid used for PC-12 transfection. Is it RNLS plasmid? Include the information about the name of plasmid used for transfection in all figure's legend. Currently it is only mentioned that PC12 cells transfected with plasmid.    

Response:

Thank you for this clarification. We have updated the Materials and Methods section to specify that an RNLS overexpression plasmid (synthesized by Genechem, Shanghai) and a corresponding empty vector control were used for PC12 cell transfection. All relevant figure legends have also been revised.

Line 179, Kindly include the source of Bovine serum albumin.    

Response:

Thank you for this suggestion. The source of Bovine Serum Albumin has been added to the Materials and Methods section.

Line 194, Kindly include the information for HPLC model used.   

Response:

Thank you for this suggestion. Information regarding the HPLC model used for the analysis has been included in the Materials and Methods section.

Line 208, Kindly include the instrument used for absorbance and also include the model information of the instrument.    

Response:

Thank you for this suggestion. The model information for the instrument used to measure absorbance in the CCK-8 assay has been added to the Materials and Methods section.

Figure-2, Legend for Figure-2 is completely wrong. Kindly include the correct legend with the experiments performed on rats. 

Response:

Thank you for this suggestion. We have corrected the legend for Figure 2 to accurately describe the in vivo data from our rat model experiments.

In Figure 3, Renalase mRNA expression is higher at 48 hours (Figure 3C), while Renalase protein expression is higher at 72 hours (Figure 3A). Could you explain why there is a discrepancy between the mRNA and protein expression levels?"    

Response:

Thanks for your kindly question. This is an astute observation. The peak of RNLS mRNA expression at 48 h preceding the peak of protein expression at 72 h is a typical biological phenomenon. There is often a temporal delay between transcription and the subsequent translation and accumulation of stable protein. Notably, if the time interval for this observation is setup more closer, the delay may not that much, but there will increase the work load. Anyway, these authors appreciate your insightful point.

In Figure-5, Changes of α-Syn content at different time points in PC12 cells transfected with? Please include transfected with which plasmid. Is it alpha-synuclein?   

Response:

We apologize for the lack of clarity. The PC12 cells in the experiments shown in Figure 5were transfected with an RNLS overexpression plasmid (or an empty vector control), not an α-Synuclein plasmid. The legend for Figure 5and other relevant figures have been revised to clearly state that cells were transfected with the RNLS overexpression plasmid. Thanks again.

Kindly extend the conclusion description since the current conclusion is very limited.

Response:

Thank you for this suggestion. We have extended the Conclusion section to better summarize the key findings.

Round 2

Reviewer 1 Report

Comments and Suggestions for Authors

The authors significantly improved the manuscript and answered my major critical points. Some of their answers should be included into the text. I mean their explanation of the presence of renalase as a 44 kDa-protein reacting with the anti-renalase antibodies due to post-translational modifications. Currently, the problem of protein identification is easily solved by mass spectrometry, but in the context of this paper, analysis of possible reasons for the molecular mass discrepancy  would be reasonable, 

Author Response

Comments and Suggestions for Authors

The authors significantly improved the manuscript and answered my major critical points. Some of their answers should be included into the text. I mean their explanation of the presence of renalase as a 44 kDa-protein reacting with the anti-renalase antibodies due to post-translational modifications. Currently, the problem of protein identification is easily solved by mass spectrometry, but in the context of this paper, analysis of possible reasons for the molecular mass discrepancy would be reasonable.

Response: Thanks and appreciates for your kindly suggestion and evaluation to our responses to your comments and one small paragraph was added at the end of in Western blot section of methodology to briefly explain the presence of renalase as a 44 kDa-protein reacting with the anti-renalase antibody due to post-translational modification.

Reviewer 2 Report

Comments and Suggestions for Authors

Thank you for adapting to my comments and taking them into account. I have no more coments. I wish you good luck in your future research:)

Line 132 please add space between: "min.After"

Author Response

Reviewer 2 Comments and Suggestions for Authors

Thank you for adapting to my comments and taking them into account. I have no more comments. I wish you good luck in your future research.

Response: Thanks and appreciates for your satisfaction to our responses to your insightful comments and suggestion.

Line 132 please add space between: "min. After"

Response: Thanks and it was fixed.